# Development of an Efficient Protocol to Obtain Transgenic Coffee, *Coffea arabica* L., Expressing the Cry10Aa Toxin of *Bacillus thuringiensis*

**DOI:** 10.3390/ijms20215334

**Published:** 2019-10-26

**Authors:** Eliana Valencia-Lozano, José L. Cabrera-Ponce, Miguel A. Gómez-Lim, Jorge E. Ibarra

**Affiliations:** 1Departamanto de Biotecnología y Bioquímica, Centro de Investigación y de Estudios Avanzados del IPN, Unidad Irapuato, Irapuato 36824, Mexico; eliana.valencia@cinvestav.mx; 2Departamento de Ingeniería Genética, Centro de Investigación y de Estudios Avanzados del IPN, Unidad Irapuato, Irapuato 36824, Mexico; jlcabre@yahoo.com.mx (J.L.C.-P.); miguel.gomez@cinvestav.mx (M.A.G.-L.)

**Keywords:** coffee, *Coffea arabica*, somatic embryo maturation, genetic transformation, cry10Aa, *Hypothenemus hampei*

## Abstract

This report presents an efficient protocol of the stable genetic transformation of coffee plants expressing the Cry10Aa protein of *Bacillus thuringiensis*. Embryogenic cell lines with a high potential of propagation, somatic embryo maturation, and germination were used. Gene expression analysis of cytokinin signaling, homedomains, auxin responsive factor, and the master regulators of somatic embryogenesis genes involved in somatic embryo maturation were evaluated. Plasmid pMDC85 containing the *cry10Aa* gene was introduced into a Typica cultivar of *C. arabica* L. by biobalistic transformation. Transformation efficiency of 16.7% was achieved, according to the number of embryogenic aggregates and transgenic lines developed. Stable transformation was proven by hygromycin-resistant embryogenic lines, green fluorescent protein (GFP) expression, quantitative analyses of Cry10Aa by mass spectrometry, Western blot, ELISA, and Southern blot analyses. Cry10Aa showed variable expression levels in somatic embryos and the leaf tissue of transgenic plants, ranging from 76% to 90% of coverage of the protein by mass spectrometry and from 3.25 to 13.88 μg/g fresh tissue, with ELISA. qPCR-based 2^−ΔΔ*C*t^ trials revealed high transcription levels of *cry10Aa* in somatic embryos and leaf tissue. This is the first report about the stable transformation and expression of the Cry10Aa protein in coffee plants with the potential for controlling the coffee berry borer.

## 1. Introduction

Coffee (*Coffea arabica* L. and *C. canephora* Pierre) is the most valuable tropical export crop worldwide, with an annual retail value of about US $90 billion. Its prices have increased by 160% during the last years [1]. *C. arabica* is highly affected by diseases and insect pests, being the coffee berry borer (CBB), *Hypothenemus hampei* (Ferrari) (Coleoptera: Curculionidae: Scolytinae), one of the major pests in the world. Larvae feed within the seeds, which are the marketable product, causing losses exceeding US $500 million annually, and affecting more than 25 million rural households involved in coffee production worldwide [2], with devastating economic consequences for farmers [3,4,5]. The cryptic habit of CBB larvae into coffee seeds can result in crop losses of up to 80%, mostly because it feeds on immature coffee berries. Unfortunately, chemical control (i.e., endosulfan) and some biological control agents, such as the white muscardine fungus, can be applied only on the surface of the fruit and act only on the adult weevil stage [6,7,8,9,10].

*Bacillus thuringiensis* (Bt) has contributed globally to insect pest control since the 1960s [11]. Currently, more than 800 sequences of Cry proteins are registered, which are grouped into 78 different classes and are specifically active mostly against some insects and nematodes [12]. The genes encoding the insecticidal proteins in some Bt microbial products have been successfully cloned, integrated, and expressed in genetically modified plants [13,14,15,16,17,18,19] to confer resistance against insect damage. Bt-protected crops such as corn, cotton, soybean, and potato have demonstrated significant benefits since their introduction in 1996. These materials provide a protection level against insects that is generally superior compared to conventional chemical pesticides. As a result, Bt crops require fewer applications of synthetic pesticides, if any. Thus, they can significantly reduce the overall use of chemical products used in pest control while preserving the population of beneficial insects [20,21].

One of the bottlenecks for an efficient plant transformation is the in vitro techniques required to obtain suitable plant tissues. Most genetic transformation protocols are based on the integration of the gene(s) of interest into the plant genome in undifferentiated plant tissues, such as the somatic embryos (SE). The development and maturing of somatic embryos are stimulated when cultured under stress conditions, such as heat, nutrient depletion, solute-based water stress, or increased levels of the plant hormone abscisic acid (ABA), whether added exogenously or induced endogenously [22]. Cytokinin signaling plays a critical role during root and stem cell establishment, allowing the root apical meristem (RAM) system initiation in SE [23,24]. Also, ABA, ethylene, light stress, MAPK cascade, and glucose signaling are involved. Understanding the role of cytokinins and analyzing the interaction of the genes involved in the two-component signaling system AHK1 and AHK3, homeodomains WUSCHEL and WOX5, ARF5 (monopteros), and morphogenetic regulators of somatic embryogenesis, such as BBM, LEC1, FUS3, and AGL15, is crucial in the development of efficient genetic transformation protocols—especially in plants with low efficiency of transformation, such as coffee.

Yet, some accomplishments have been achieved. Transgenic coffee plants expressing Cry proteins were first developed by Leroy et al. [16] and further analyzed under field conditions by Perthuis et al. [25]. These plants express the Cry1Ac protein that confers resistance to the coffee leaf miner, *Leucoptera coffeella* (Lepidoptera: Lyonetiidae). Although several characterized Cry toxins are active against lepidopteran insects, far fewer Cry proteins present toxicity to coleopteran species [26,27,28]. Méndez-López et al. [29] demonstrated that *B. thuringiensis* serovar *israelensis* (Bti), which contains the Cry10Aa protein, showed high toxicity levels against the CBB. Later, specific and high activity of Cry10Aa toward the cotton boll weevil (CBW), *Anthonomus grandis* Boheman, was proved in vitro [30], and in transgenic cotton plants under greenhouse conditions, showing high levels of toxicity against the CBW [17]. In this report, we describe the first stable and efficient genetic transformation of *C. arabica* var. Typica using particle bombardment, with high levels of germination and transformation efficiency, which provided constitutive high expression levels of the Cry10Aa δ-endotoxin.

## 2. Results

### 2.1. Cry10aa Codon Optimization for Coffee C. arabica Genetic Transformation

The modification of the *cry10Aa* gene sequence described here was focused on the optimization of codon usage in coffee. We altered 73.6% of the codons in the coding region of the *cry10Aa* gene based on our survey of codon usage in caffeine and theobromine in *C. arabica* genes, which are highly expressed in seeds. This corresponds to 40.3% frequencies of the terminal nucleotide of leucine, valine, aspartic acid, and isoleucine, and 33.3% specific changes in serine, threonine, and proline (Appendix A). The resulting protein structure was compared with the original one, and a prediction of the Cry10Aa three-dimensional model revealed, as expected, in which the optimized Cry10Aa toxin showed its original structure. This prediction suggested that codon optimization had no effect on the structural conformation of the modified Cry10Aa δ-endotoxin, and that it could be functional.

### 2.2. Generation of Plasmid pMDC85/Cry10Aa

The synthesized *cry10Aa* gene (1993 bp), cloned in pUC57, was amplified and cloned into pCR8/GW/TOPO (4792 bp), which was confirmed by amplification and digestion of the construct with EcoRI, to obtain fragments of 2799 bp, 1386 bp, and 607 bp. When sub-cloned in the plant transformation vector pMDC85 (12,462 bp), amplification and digestion with BsaI confirmed the presence of 5526 bp, 4592 bp, 1993 bp, and 727 bp fragments, where the 1993 bp fragment corresponded to the *cry10Aa* gene insert in pMDC85. Consequently, the new construct pMDC85/*cry10Aa* was used for genetic transformation of coffee by biobalistics (Figure 1).

### 2.3. Induction of Somatic Embryogenesis

Indirect somatic embryogenesis was induced from leaf explants of *C. arabica* var. Typica. Three months were required to induce the development of somatic embryogenesis in order to be ready for stable genetic transformation. When leaf explants of coffee cultured in callus induction medium (PEM medium), as described by Van Boxtel et al. [31], pro-embryogenic masses were produced after one month with 93% efficiency of cultivated explants. Somatic embryo development was achieved when pro-embryogenic masses were sub-cultured in SE-P medium after two months in culture, with an efficiency ranging from 82% to 100%. Embryogenic cell lines with a high (90%) efficiency of conversion from a mature somatic embryo to plant were chosen for genetic transformation experiments.

### 2.4. Somatic Embryo Maturation

Two treatments were evaluated to induce somatic embryo maturation. Evaluation was made according to the rate of germination and complete plant regeneration in subsequent experiments. Medium supplemented with 9 g/L gelrite induced 95% maturation as compared to 40% with 3 g/L gelrite. It is important to notice that an adequate separation of single somatic embryos is required to achieve somatic embryo maturation (Figure 2C). In the first month of culture, secondary somatic embryos developed under cytokinin and osmotic stress. When separated individually, successful somatic embryo maturation occurred (Figure 2D,E).

### 2.5. Selection of Transgenic Clones

Plasmid pMDC85/*cry10Aa* was used to genetically transform *C. arabica* globular somatic embryos (Figure 2A). Five osmotic conditions were evaluated before bombardment, and after bombardment, when selection was carried out by using 50 mg/L hygromycin, under dark conditions to produced transgenic cell lines of coffee. Four sub-cultures were made every two weeks until negative controls (non-bombarded) ceased cell division, and bombarded explants started to produce resistant embryogenic structures. All evaluated treatments produced transgenic cell lines using this selection method. As expected, osmotically treated SE prior to bombardment produced the higher amount of transgenic lines, as shown in Table 1. Interestingly, the sucrose 6% treatment showed the highest transformation efficiency and number of grown plants. The negative control (CP2 = SE-P) showed the lowest levels. Non-bombarded plates showed no growth in the selective medium.

In total, 835 different genetic transformation events were produced, from which 17 events were selected based on their evident ability of propagation in hygromicin-containing medium and green fluorescent protein (GFP) expression for the further evaluation of Cry10Aa expression. From the 17 selected events, seven transgenic lines were derived from sucrose 6%, four from 0.15 M mannitol + 0.5 M sorbitol, two from 0.3 M mannitol + 0.3 M sorbitol, two from sucrose 12%, and two from no osmotic treatment. Stable GFP expression was observed in 100% of the 835 transgenic cell lines after the sixth sub-culture (after three months) (Figure 2F,G) in hygromycin-containing medium (Figure 2B).

### 2.6. Regeneration of Transgenic Plants

The selected 17 hygromycin-resistant embryogenic cell lines were propagated for three months, subjected to somatic embryo maturation as described above, and sub-cultured to plant regeneration conditions on SE-G medium. The germination of somatic embryos occurred when they were sub-cultured on SE-G medium for two months. The size of cotyledons in mature somatic embryos varied in size: 10% large, 60% medium, and 30% small. 95% of transgenic cell lines were able to regenerate normal plants after 10 weeks, which were established in soil under growth chamber conditions. All plants exhibited normal growth and phenotypes as compared to those of WT coffee plants subjected to no selection conditions (Figure 2H).

### 2.7. PCR and qPCR Analysis of Cry10Aa

qPCR analysis was performed to detect the presence of the *cry10Aa* gene in the DNA of 17 putatively transformed plants derived from independent transformation events, and to assess the level of expression of each event, using one plant per transformation event. The predicted 201-bp amplicon that indicated the presence of the *cry10Aa* gene was detected in 12 of the transgenic plants analyzed, indicating the high efficacy of hygromycin selection. Such an amplicon was not observed in wild-type (WT) plants. 

After normalization based on the expression of plant endogenous genes (actine, RP29, and 24S), qPCR analysis of *cry10Aa* transcript relative expression (Log2) in the leaves of the 17 transgenic plants and two wild-type control plants was carried out. Notably, event 2 (E2) presented the highest *cry10Aa* transcript expression level (4.56-fold), whereas other events (E1, E4, E8, E11, E12) presented the lowest expression levels (0.11, 0.5, 0.78, 0.17, 0.21-fold, respectively); eight events ranged from 3.22 to 3.87-fold; and no expression was observed in the WT plants (Figure 3).

### 2.8. Southern Blot and Hybridization Analysis

Four regenerated hygromycin-resistant and *cry10Aa* positive lines were analyzed by Southern blot hybridization to confirm the presence of the 607 bp and 1386 bp EcoRI fragments generated by the digestion of genomic DNA of coffee. A biotinylated probe of 1993 bp of *cry10Aa* was used for hybridization. Different hybridization patterns were found among the four positive lines, while no signal was present in non-transgenic coffee plants (Figure 4).

A hybridization signal of 607 bp corresponding to the 3′–end of the *cry10Aa* gene was observed in events E2, E3, and E4, but was absent in E1. The second expected signal of 1386 bp corresponding to the rest of *cry10Aa* gene was clearly found only in the E2 event. 

The presence of positive hybridization signals of the predicted size shows that the plants contain at least one intact copy of the expected fragment.

In all transgenic lines, bands with molecular weights different from expected were found. This indicates either that multiple independent insertions occurred, or that the integrated fragments are long tandem repeats resulting from re-arrangements. 

All the transgenic clones tested displayed unique hybridization patterns, indicating that these transgenic plants were derived from independent transformation events. This integration pattern correlates with qPCR and immunodetection of the Cry10Aa protein, where event E2 has the highest relative expression, and produced more Bt proteins as compared with the remaining events.

### 2.9. Immunodetection of the Cry10Aa Protein 

ELISA and Western blot analyses of Cry10Aa expression levels from somatic embryos and leaves varied among the four selected transgenic coffee plants, although a pattern was observed. First, quantification by ELISA showed that the relative expression in leaves was lower (3.25 to 7.6 μg/g fresh weight leaf) than the expression detected in somatic embryos (4.6 to 13.88 μg/g somatic fresh weight) (Figure 5). No detection was observed in WT plants. Western blot analysis of total extract pre-purified proteins from both leaves and embryos was performed from SDS-PAGE transfers. The presence of the Cry10Aa active protein with approximately 77 kDa is observed in events E2, E3, and E4 (Figure 6A,B). For mass spectrometric analysis, a band from SDS-PAGE corresponding to the Cry10Aa protein was analyzed. Results showed a score of 8680.46, and 16 peptides showing 100% identity with the reported protein, which completed 76% coverage of the protein (Figure 6C). This result identified the Cry10Aa protein in the analyzed tissues, which was absent in the extracts from WT plants.

## 3. Discussion

Somatic embryogenesis (SEG) induced from leaf explants has been the most widely used target tissue in coffee genetic transformation [32]. Published protocols have shown that SEG induction and proliferation is time consuming, ranging from 9 to 15 months until they can be used for stable genetic transformation [16,33,34]. In most of these publications, SE conversion to plants in *C. arabica* has proved to be more difficult and time consuming than in *C. canephora.* As noted by several authors, auxins negatively affect SE development in coffee [35,36,37,38,39]. Cytokinin signaling has demonstrated that it plays a critical role during root and stem cell niche formation, allowing RAM system initiation in SE [23,24]. To improve the SE maturation in *C. arabica* var. Typica, we hypothesize that SE under osmotic stress in the presence of cytokinins and genes involved in a two-component signaling system will activate key factors such as Wuschel and WOX5, ARF5 (Monopteros), as well as morphogenetic regulators of somatic embryogenesis such as Baby Boom, LEC1, FUS3, and AGL15, to develop mature SE. The results of gene expression analysis will be published elsewhere.

Osmotic stress was activated by using 9 g/L gelrite. This may explain why 95% of treated SE under osmotic stress was able to germinate and develop into plants in contrast with 40% in untreated SE. In higher plants, seeds underwent desiccation during later stages of zygotic embryo development; this is a process that plays an important role in the transition between embryo maturation and germination. A similar process is critical in somatic embryogenesis induced under in vitro conditions. SE maturation is a complex process that is influenced by many factors, including abscisic acid (ABA) coupled with the osmotic water potential of the medium. SE maturation has been induced in several species by polyethylene glycol (PEG)-inducing osmotic stress such as *Hevea brasiliensis* [40], *Picea bies* [41], *Glycine max* [42], *Aesculus hippocastanum* [43], and *Carica papaya* [44,45,46]. ABA and PEG treatment has been used in combination to induce SE maturation in spruce and larch species [47,48]. In addition, partial desiccation treatments promoted plantlet development [49]. Exogenous ABA addition has been used in Persian walnut [50]. The maturation of SE of various species is routinely promoted by using media with gelrite concentrations exceeding the standard 3 g/L up to 12 g/L in several species of *Pinus* [51], avocado [52,53,54], and hybrid larch [55]. We were able to reach up to 95% germination and total plant regeneration with SE under osmotic stress treatment in contrast with 40% in untreated SE.

Overall, it takes at least 12 months to produce regenerated transgenic plants in soil conditions, but it may reach up to 18 to 27 months, as reported in other publications [16,31,32,56]. Our efficiency of transformation was estimated at 25.6% of stable genetic transformation, which is high compared to other publications: 1% by Leroy et al. [16], 12.5% by Ribas et al. [56]; four putative transformation events per 50 mg of bombarded explants in Albuquerque et al. [33]; and 22.8% by Breitler et al. [34]. In this work, the efficiency was calculated on the number of hygromicin-resistant cell lines per number of embryogenic aggregates.

The protocol reported here was successfully used to genetically transform coffee SE, mediated by particle bombardment, to integrate the *cry10Aa* gene found in Bti. The activity of the Cry10Aa toxin on the CBB was evaluated previously [57], demonstrating that Bti, known mainly by its mosquitocidal activity, was highly toxic to CBB first instar larvae. Its activity was also corroborated on another related coleopteran pest, the CBW [29], and its expression in a baculovirus system showed an LC_50_ of 7.12 μg/mL, and 6.35 μg/mL in transformed cotton plants [17,30]. Our transformed coffee plant showed different Cry10Aa expression levels of 7.6 to 13.88 μg/g of fresh tissues of leaves and embryos, respectively. Variability in the toxicity of Bt cotton has been reported by Ma et al. [58]. The reduction in Cry toxin biosynthesis in late-season cotton tissues could be attributed to the overexpression of the *cry* gene in earlier stages, which leads to gene regulation at post-transcription levels as well as to transgene silencing during later stages [17,59,60].

## 4. Materials and Methods

### 4.1. Cry10aa Gene Codon Optimization for C. arabica 

The *cry10Aa* gene sequence from the *B. thuringiensis* serovar *israelensis* plasmid pBtoxis (NC_O10076.1 NCBI) contains a total of 2028 bp, which code for a protein of 675 amino acids (WP_001070502). This sequence was analyzed, and its codon usage in a plant system was optimized with the Genscript and Integrated DNA technologies softwares (www.genscript.com; https://www.idtdna.com). The consensus region was selected and verified by in silico prediction, to evaluate the absence of conformational changes of the protein, using the three-dimensional prediction model with the I-Tasser software [61].

### 4.2. Cry10Aa Expression Vector

The codon-optimized *cry10Aa* gene sequence was chemically synthesized by GenScript, Inc. (Piscataway, NJ, USA) and cloned into the pUC57 vector, sub-cloned in PCR/TOPO8GW (Invitrogen, Carlsbad, CA, USA), and recombined in the gateway plasmid pMDC85 with LR clonase™ II Enzyme (Invitrogen, Carlsbad, CA, USA) Mix [62]. The resulting plasmid (pMDC85/*cry10A*) contains the hygromycin phosphotransferase (*hpt*) gene that confers resistance to hygromycin B as a selecting marker, the mGFP (green fluorescent protein) united to a nucleus signal peptide as reporter gene, and the *cry10Aa* gene under the 2X35S promoter and NOS terminator.

### 4.3. Induction of Somatic Embryogenesis

Somatic embryogenesis of *C. arabica* var. Typica (INIFAP-Tapachula, Chiapas, Mexico), was induced from leaf explants, derived from eight-month-old trees. Explants were disinfected by gasification in a vacuum chamber with a mixture of 50 mL of sodium hypochlorite from a commercial bleach (1.2% active chlorine) and 50 mL of HCl 6N for 15 min and washed four times with sterile distilled water [63]. Disinfected leaves were cut into pieces of 1 cm^2^ and cultured on the callus induction PEM (CP) medium as described before [31]: half-strength MS medium [64], 30 mg/L sucrose, 100 mg/L casein hydrolysate, 400 mg/L malt extract, 10 mg/L thiamine, 1 mg/L nicotinic acid, 1 mg/L pyridoxine, 1 mg/L glycine, 100 mg/L myo-inositol, 0.5 mg/L 2.4-D, 1 mg/L indole-3-butyric acid (IBA), 2 mg/L 2-isopentenyladenine (2-iP), and solidified with 2.4 g/L gelrite. pH was adjusted to 5.8 before autoclaving. After two months in the dark, pro-embryogenic masses were sub-cultured into a SE-P medium (CP2), as described earlier [31], for somatic embryogenesis induction with slight modifications: half-strength MS salts medium [64], 30 mg/L sucrose, 200 mg/L casein hydrolysate, 800 mg/L malt extract, 60 mg/L adenine free-base, 1 mg/L 2.4-D, and 4 mg/L BAP, pH 8.0 and solidified with 3.2 g/L gelrite.

### 4.4. Somatic Embryo Maturation of C. arabica

To evaluate the role of cytokinin signaling in somatic embryo maturation, two treatments were evaluated: (a) SE-M3 with MS medium supplemented with 0.2 mg/L BAP, 0.1 mg/L kinetin, 1% glucose, 3.0 g/L gelrite (−0.49 MPa), pH 5.8; and (b) SE-M9, with MS medium supplemented with 0.2 mg/L BAP, 0.1 mg/L kinetin, 1% glucose, 9.0 g/L gelrite (−1.47 MPa), pH 5.8. Plates containing somatic embryos were incubated at 25 ± 2 °C, under a 12/12 h photoperiod at 50 µmol/m^−2^ s^−1^ irradiance provided by fluorescent lamps T8 Phillips P32T8/TL850 mixed with natural light increasing red/far red light in the spectrum (Appendix A). The first sub-culture consisted in aggregated somatic embryos (about 20) from the globular to the early torpedo stage, and then separated and individually sub-cultured for one month until the cotyledonary stage developed.

### 4.5. Isolation of RNA and qPCR Analysis

Total RNA from somatic embryos in the maturation stage was isolated using Trizol (Invitrogen, Carlsbad, CA, USA), while RNA concentration was measured by its absorbance at 260 nm, the ratio 260 nm/280 nm was assessed, and its integrity was verified by electrophoresis in agarose 2% (*w*/*v*) gels. Samples of cDNA were amplified by PCR using SYBR Green qPCR (Bio-Rad, Hercules, CA, USA) in Real-Time PCR Systems (CFX Bio-Rad, Hercules, CA, USA). The expression of actin, RP29, and S24 was used as reference for calculating the relative amount of target gene expression using the 2^−ΔΔ*C*t^ method [65]. qPCR analysis was based on at least three biological replicates for each sample with three technical replicates.

### 4.6. Preparation of the DNA Plasmid Construct (pMDC85/Cry10Aa)

Plasmid pMDC85/*cry10Aa* was amplified in *Escherichia coli,* isolated by alkaline lysis [66], and purified with the Invitrogen DNA Purification Kit (Invitrogen, Carlsbad, CA, USA). Purified DNA was resuspended in TE buffer (1 mM Tris pH 7.8, 0.1 mM Na_2_EDTA) and adjusted to 1 mg/mL.

### 4.7. Osmotic Treatment

Embryogenic aggregates at the globular stage were propagated and selected one week before genetic transformation. Each bombardment plate contained 70 mg of fresh weight (~25 aggregates with 20 embryos each) placed in the center of the plate. Four different osmotic-conditioning media were used: 0.3 M sucrose (−0.869 MPa.), 0.6 M sucrose (−1739 MPa), 0.15 M mannitol + 0.15 M sorbitol (0.801 MPa), and 0.3 M mannitol + 0.3 M sorbitol (1602 MPa) separately in the solidified medium (SE-P) used for pre- and post-bombardment culture of somatic embryos [67,68]. The osmotic treatment consisted of two steps: a 24-h treatment prior to bombardment, and a 24-h treatment after bombardment.

### 4.8. Particle Bombardment

Plasmid pMDC85/*cry10Aa* was precipitated onto gold particles (1.6 μm) as described before [69] and later modified [70]. Prior to bombardment, DNA-coated microprojectiles were resuspended in 400 μL of absolute ethanol. Aliquots of 10 μL (1.66 μg DNA associated with 125 μg of gold particles) were delivered to each macrocarrier membrane, air-dried to remove ethanol, and bombarded onto somatic embryos at the globular stage, using the helium-driven version of PDS-1000 [71]. The gap distance between the rupture membrane and the flying disk was 1.2 cm, the macrocarrier (a kapton disk) traveled 1.2 cm before impact with a steel-stopping screen. Fifty plates were bombarded, each containing SE-P solidified medium, which was used for bombardment experiments. The bombardment chamber was evacuated to 0.07 atmospheres, and the gas acceleration tube was pressurized with the chosen helium gas pressure. Target tissues were placed 7.0 cm from the launch point and bombarded once at 900 psi.

### 4.9. Selection of Transgenic Embryogenic Lines

Three days following bombardment, somatic embryos were subjected to selection in SE-P medium supplemented with hygromycin (50 mg/L). Sub-cultures were made every two weeks, six times, until developing resistant somatic embryos were observed. Hygromycin-resistant cell lines, containing aggregates of at least five somatic embryos coming from a single somatic embryo, were detached and sub-cultured separately in SE-P medium with hygromycin (50 mg/L) to propagate them. The transformation efficiency was measured in terms of the proportion of embryogenic calli (20 per plate) that grew on hygromycin medium after bombardment.

### 4.10. Transgenic Somatic Embryo Maturation

Transgenic somatic embryo maturation was induced in SE-M9: MS medium supplemented with 0.2 mg/L BAP, 0.1 mg/L kinetin, 1% glucose, 9.0 g/L gelrite (−1.47 MPa), pH 5.8. Incubation parameters were similar to those mentioned in somatic embryo maturation.

### 4.11. Regeneration of Transgenic Plants

Mature somatic embryos were sub-cultured in germination SE-G medium: MS medium, 30 g/L sucrose, 1 g/L activated charcoal, 3 g/L gelrite, and karrikins derived from smoke-water (0.5%) [72] and incubated at 25 ± 2 °C for one month under the same illumination conditions mentioned in somatic embryo maturation. Then, they were sub-cultured in flasks with SE-G medium until plants reached 10 leaves and 27 cm root length. Regenerated plants were potted in vermiculite and soil (1:1 *v*/*v*), incubated in a plant growth chamber at 50 µmol/m^−2^ s^−1^ irradiance provided by fluorescent T8 Phillips P32T8/TL850 lamps (Beijing, China) with a photoperiod of 16/8 h and temperature of 21 ± 2°C.

### 4.12. Analysis of GFP Expression

Selected embryogenic structures were evaluated for the transient and stable expression of mGFP 5, 15, and 30 days after bombardment. Somatic embryos were screened for GFP expression using a Leica MZ Fluo III (optic 0.63 Zeiss) fluorescence microscope supplied with a DC 300F camera (Leica Microsystems, Welzlar, Germany) and a plant GFP filter no. 3 from Leica (excitation wavelengths 470–540 nm, emission wavelengths 525–550 nm). Autofluorescence from chlorophyll was blocked using a red interference filter and propidium iodide staining.

### 4.13. Southern Blot Analysis

Total genomic DNA was isolated from the leaf tissue of putative transgenic and wild-type plants, as described above. Aliquots of 20 mg of genomic DNA were digested with EcoRI (10 U/mg), which fragmented the *cry10Aa* sequence into two bands (607 pb and 1386 pb). Then, products were electrophoresed in a 1.0% agarose gel and transferred to positively charged nylon membranes (Hybond-N+, Amersham Pharmacia Biotech, Little Chalfont, UK) using 2X SC, as described earlier [73]. Membranes were prehybridized for 24 h at 60 °C in 2X SCP, 0.5% BSA, and hybridized overnight at 60 °C [73]. A 1993 bp fragment (amplified with primers CRY10F and CRY10R) that contains the whole sequence of the *cry10Aa* was used as a probe, which was labeled with biotin-DNA. The hybridized biotin-labeled probes were detected with a streptavidin antibody conjugated with alkaline phosphatase (AP) and revealed using an AP conjugate substrate kit (ThermoFisher, Vilnius, Lithuania).

### 4.14. Immunodetection of the Cry10Aa Protein

For ELISA immunodetection and quantification of the Cry10Aa protein in transgenic and wild-type plants, 60 mg of leaves and transgenic somatic embryos tissue were transferred to Eppendorf tubes and homogenized in 1.2 mL of citrate buffer (pH 5) mixed with 0.05% Viscozyme^®^ L-Sigma. Then, total protein extracts were stirred for 45 min at 37 °C. Supernatants were collected after centrifugation at 9000× *g* for 15 min at 4 °C. An entire 96-well microtiter plate was coated with 5 μg of total protein extracts, either from putative transgenic or WT plants. ELISA microplate kits from Envirologix (Portland, OR, USA) were incubated for 1 h at 37 °C, and subsequently, non-specific binding sites were blocked by incubation in 3% BSA dissolved in buffer phosphate-buffered saline (PBS) (100 mM NA_2_HPO_4_, 17 mM KH_2_PO_4_, 5 M NaCl, 27 mM KCl, pH 7.4) with 0.05% Tween-20 overnight at 37 °C. Next, microplate wells were incubated with Anti-Cry10Aa polyclonal antibody, produced in rabbit by GenScript (Piscataway, NJ, USA) from the synthetic peptide VSSDSKIVKGPGHT, based on the Cry10Aa amino acid sequence and in silico immunogenic studies. Anti-Cry10Aa was used as a primary antibody (1:1000 in 2.0% BSA-PBS) for 2 h at 37°C. After washing (PBS with 0.05% Tween-20), the microplate wells were incubated with anti-rabbit antiserum with alkaline phosphatase conjugate (Sigma, St Louis, MO, USA) used as a secondary antibody (1:1000 in 1.0% BSA-PBS with to 0.05% Tween-20) for 2 h at 37 °C. After washing as described above, Cry10Aa was detected in colorimetric assays by incubating the microplate wells with ABTS [2,2′-azino-bis-(3-etilbenzotiazolin-6-sulfonic)], dissolved in detection buffer (dibasic sodium phosphate, 0.1 M citric acid, pH 5.0). The absorbance of the developed color was measured at 450 nm in a Mark™ Microplate Absorbance Spectrophotometer (Bio-Rad, Hercules, CA, USA) using the Microplate Manager^®^ Software (Version 5.2.1 Build 106, Bio-Rad, Hercules, CA, USA). All data were statistically analyzed by ANOVA and Tukey’s HSD test, with 0.05 probability in the R software (version 3.2.1, R Foundation for Statistical Computing, Vienna, Austria).

For Western blot immunodetection of Cry10Aa in transgenic and WT plants, 120 mg of leaves and transgenic somatic embryos from extracted fresh tissue were homogenized in 25 mL of total protein extraction PBS with 10 mM sodium metabisulphite and 0.5% Triton X-100. After 24 h under slow stirring at 4 °C, the homogenate was centrifuged twice at 5000× *g* for 10 min at 4 °C. Aliquots of 20 mg for each total protein extract were subjected to 12% SDS-PAGE. After electrophoresis, the separated proteins were transferred from the gel to a nitrocellulose membrane (Hybond N+, Amersham Pharmacia Biotech, Little Chalfont, UK) using a semi-dry TransBlot Cell Unit (Bio-Rad, Hercules, CA, USA). The membrane was blocked overnight in 3% Svelty^®^ milk in PBS with 0.05% Tween-20 at room temperature. After washing three times in PBS–Tween-20 and once in PBS, the membrane was incubated in 2% Svelty^®^ milk in PBS with anti-Cry10Aa primary antibody (1:1000) for 24 h at room temperature. Subsequently, after a rapid wash, the membrane was incubated in anti-rabbit IgG alkaline phosphatase-conjugated (Bio-Rad, Hercules, CA, USA) secondary antibody (1:3000) for 3 h at room temperature. Finally, the Alkaline Phosphatase Detection kit (Bio-Rad, Hercules, CA, USA) was used for the colorimetric detection of Cry10Aa according to the manufacturer’s instructions.

### 4.15. Mass-Spectrometry Analysis of the Cry10Aa Protein

The protein identification of Cry10Aa in coffee transgenic plants was done by mass spectrometry [74]. Expected bands of 77 kDa for Cry10Aa were separated from SDS-PAGE electrophoresed gel for further analysis. Analysis was performed in an Agilent 6460 LC/MS/MS triple quadrupole (CINVESTAV, Irapuato, Mexico) using an Ultimate 3000 nano High-Performance Liquid Chromatography (Thermo Fisher Scientific Inc, Waltham, MA, USA). The spectrum was recorded with Xcalibur software (3.0.63) (Thermo Fisher Scientific Inc, Waltham, MA, USA). Mass spectrometry data was processed using the Trans-Proteomic Pipeline (TPP) software [75].

## 5. Conclusions

The protocol developed in this work improves the efficiency to genetically transform *C. arabica* plants by manipulating the osmotic response of the genes involved in the maturation of somatic embryos. The desiccation of SE plays an important role for the maturation and germination of SE, similar to zygotic embryogenesis. A high concentration of gelrite (osmotic stress) induced the SE maturation of *C. arabica* under in vitro conditions in the presence of cytokinins. High numbers of conversion to plants were obtained and successfully established in soil conditions successfully. The stable expression of the Cry10Aa protein in the transgenic coffee plants was proven, with expression levels comparable to other positive reports. Therefore, the transgenic coffee plants reported here may become an important control measure for such an important pest as it is the CBB, globally.

## Figures and Tables

**Figure 1 ijms-20-05334-f001:**
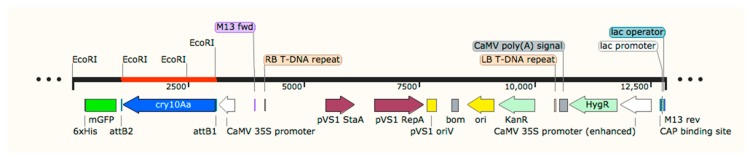
Schematic representation of the plasmid pMDC85/*cry10Aa*. The 12,838 bp-long plasmid consists of the *cry10Aa* gene under 2XCaMV35S (cauliflower mosaic virus 35S promoter with a duplicated enhancer region) and 3′NOS (nopaline synthase terminator); HygR (hygromycin phosphotransferase) gene (*hph*) under 2XCaMV35S and 3′NOS terminator, mGFP (modified green fluorescent protein) under 2XCaMV35S promoter and 3′NOS terminator. 6xHis (6x histidine affinity tag); attB1-B2 (mutant version of *att*B); RB (right border repeat from nopaline C58T-DNA); pVS1 StaA (stability protein from *Pseudomonas* plasmid pVS1); pVS1 RepA (replication protein from *Pseudomonas* plasmid pVS1); Bom (bases of mobility region from pBR322); Ori (high-copy-number ColE1/pMB1/pBR322/pUC origin of replication); KanR (aminoglycoside phosphotransferase *aphA-3*, which confers kanamycin resistance); LB T-DNA repeat (left border repeat from nopaline C58 T-DNA); CaMV poly(A) signal (cauliflower mosaic virus polyadenylation signal); CAP binding site (*E. coli* catabolite activator protein); lac promoter (three segments) promoter for the *E. coli lac* operon; lac operator (laccase repressor encoded by *lacI*); and M13 rev. (common sequencing primer).

**Figure 2 ijms-20-05334-f002:**
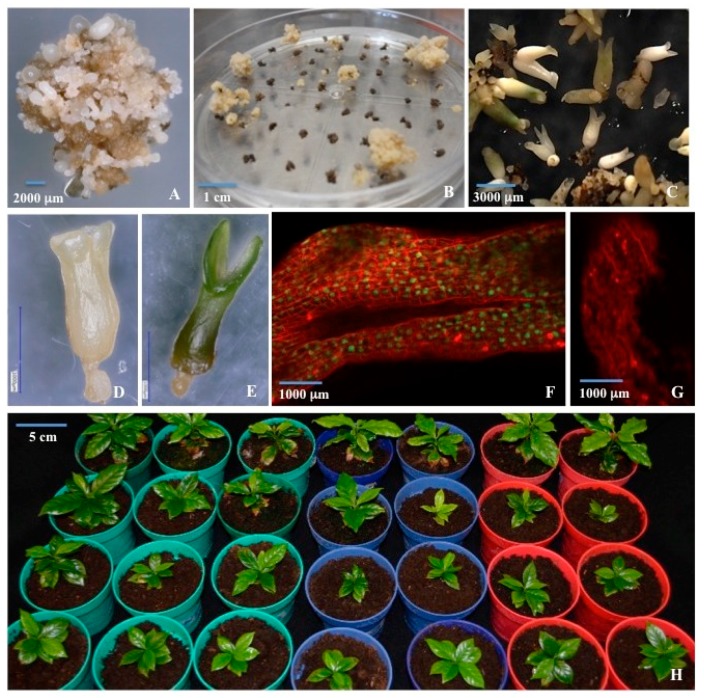
(**A**) *C. arabica* embryogenic calli induced from leaf explants used for stable genetic transformation. (**B**) Hygromycin-resistant embryogenic calli of *C. arabica* derived from bombarded globular structures after four months of selection, showing independent events of genetic transformation. (**C**) Somatic embryos under osmotic stress in the first month of maturation. (**D**,**E**) Individual somatic embryo under osmotic stress in the second month of maturation. Scale bar 1000 μm. (**F**) Green fluorescent protein (GFP) stable expression in nuclei of transgenic mature somatic embryos of *C. arabica*, after three months of growth on 50 mg/L hygromycin-containing medium. (**G**) Control somatic embryo, showing no GFP expression. (**H**) Regenerated plants of *C. arabica* after six months in soil conditions. Wild-type plants in red pots, transgenic plants in green and blue pots.

**Figure 3 ijms-20-05334-f003:**
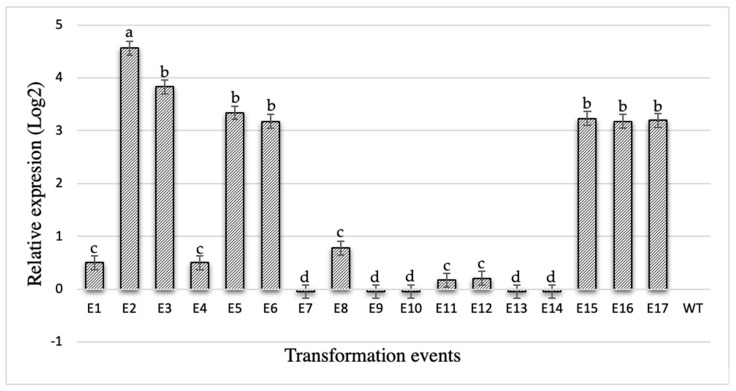
Relative expression as measured by *cry10Aa* transcript detection with quantitative PCR (qPCR) in Cry10Aa transgenic coffee plants. *cry10Aa* transcript relative expression (Log2) in leaves of 18 plants. Letters represent the level of statistical significance (LSD-test).

**Figure 4 ijms-20-05334-f004:**
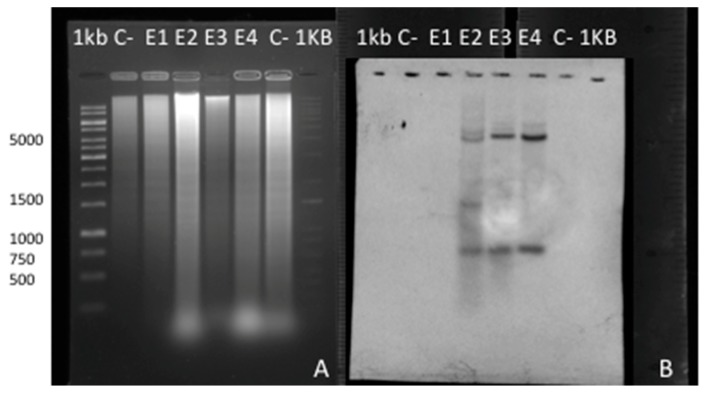
Southern blot analysis of transgenic coffee plants. (**A**) Southern blot hybridization analysis of regenerated plants of *C. arabica* digested with EcoR1. (**B**). The blot was hybridized with Biotin labeled *cry10Aa* 1996 bp fragment. Lanes E1, E2, E3, and E4 correspond to genomic DNA from three independent transformation events. C-lane: DNA extracted from an untransformed control plant.

**Figure 5 ijms-20-05334-f005:**
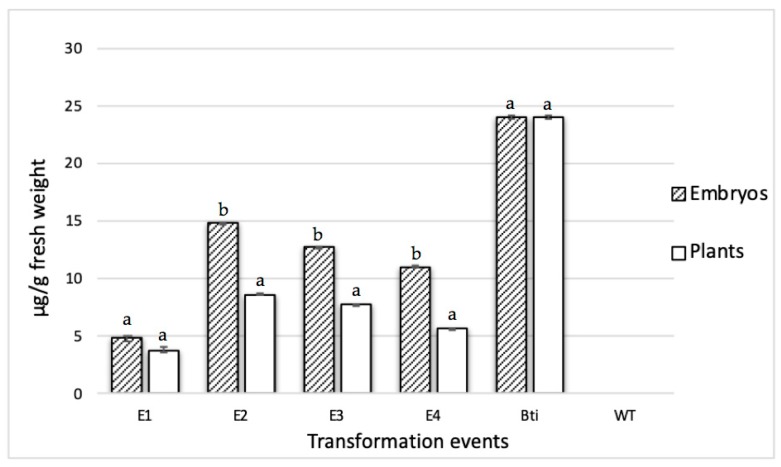
Cry10Aa protein immunodetection in transformed and wild-type coffee plants by ELISA. The primary antibody used in all experiments was the polyclonal anti-Cry10Aa. Total protein per well, 10 μg. Letters represent statistical significance (Student’s *t*-test) between embryos and leaves from each transformation event.

**Figure 6 ijms-20-05334-f006:**
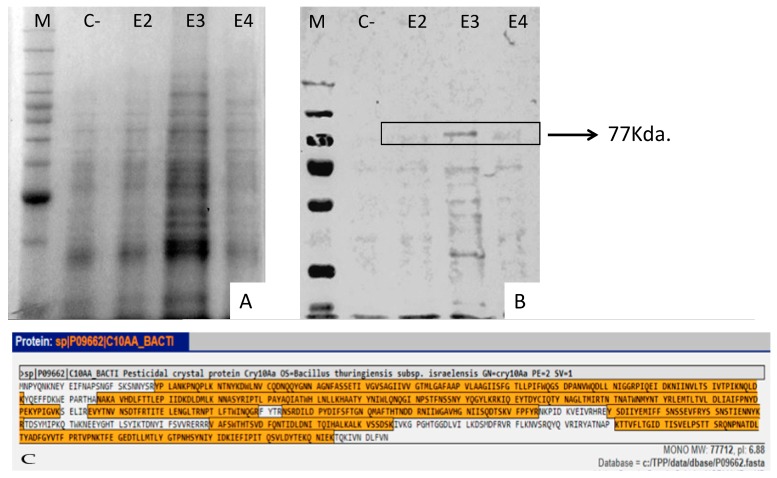
(**A**) SDS-PAGE of pre-purified proteins from plant tissues, M (molecular weight marker), C (wild-type plant), E2–E3–E4 (transformation events); (**B**) Western blot, putative Cry10Aa protein is indicated; (**C**) Mass spectrometry analysis from SDS-PAGE bands corresponding to the Cry10Aa protein (shadowed sequence fragments confirm the expression of Cry10Aa) with a coverage of 76% of the Cry10Aa protein.

**Table 1 ijms-20-05334-t001:** Summary of genetic transformation events of *C. arabica* mediated by particle bombardment of somatic embryos with plasmid pMDC85 under different osmotic conditions.

Treatment	Bombarded Plates	Transgenic Embryogenic Lines	Transformation Efficiency (%)	Transformed Events/Plate	Plants Grown (%)
CP2 (control)	50	85	8.5	1.7	98
Sucrose 6%	50	256	25.6	5.12	100
Sucrose 12%	50	133	13.3	2.56	87
Mannitol 0.15 M + Sorbitol 0.15 M	50	197	19.7	3.94	95
Mannitol 0.3 M + Sorbitol 0.3 M	50	164	16.4	3.28	80

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
