# Peer review of "Development of an Efficient Protocol to Obtain Transgenic Coffee, Coffea arabica L., Expressing the Cry10Aa Toxin of Bacillus thuringiensis"

_ijms, 2019, doi:10.3390/ijms20215334_

Round 1

Reviewer 1 Report

My main concern is about the absence of experiments to test the activity of the transgenic plants towards the coleopteran pest,  the coffee berry borer (CBB), Hypothenemus hampei (Ferrari) (Coleoptera: Curculionidae: Scolytinae). Although one would expect that if the plant is expressing the Cry10Aa endotoxin it will help to control this pest (or other susceptible ones), I think it is really necessary, to perform the corresponding bioassays. In fact E2 is stated  to be the one expressing the highest amount of protein but then it is not detected at the western blot experiment, at least from the images which have been included.

Other concerns are referring to the text,

I don't see the point of including a prediction of a protein structure to confirm that it has the same structure of an identical protein: after codon optimization the aa sequence is the same so it is evident.  The experiments analysing expression of genes involved in cytokinin signaling, homedomains, auxin signaling,  under non-osmotic (normal) conditions and osmotic stress conditions can be a "plus" but  I either don't see the point of including such an analysis in a paper where the main point is to get plant transformants in a easiest way. So, I think is quite a lot of work , and quite a lot of introduction ,  discussion and references focused on this matter and as I said before the objective of the paper should be the protocol of transformation and the effectivity of the plant for pest control.

Minor concerns:

Figure 6: define panels A and B in figure legend

Figure 7, spelling: transformation, not trasnformation

Figure 8: include a control with purified Cry10Aa

Author Response

My main concern is about the absence of experiments to test the activity of the transgenic plants towards the coleopteran pest, the coffee berry borer (CBB), Hypothenemus hampei (Ferrari) (Coleoptera: Curculionidae: Scolytinae). Although one would expect that if the plant is expressing the Cry10Aa endotoxin it will help to control this pest (or other susceptible ones), I think it is really necessary, to perform the corresponding bioassays.

Answer: The importance of this MS is focused on the high transformation efficiency we obtained and the confirmation of the stable expression of the transgene reported here, as earlier reports show lower transformation efficiency and regeneration of coffee plants. We understand that the MS would be significantly improved with the addition of efficiency data against CBB; however, as you know, coffee is a perennial plant, meaning that the first flowering occurs after 4 to 5 years after plantation. At this moment, we have transformed coffee trees about 1.5 meters tall and hundreds of smaller plants, and we expect to have the first flowering within 1 year. We expect to have an MS ready, about their insecticidal activity, within 2 years.

In fact E2 is stated to be the one expressing the highest amount of protein but then it is not detected at the western blot experiment, at least from the images which have been included.

Answer: Agree. The detection of a band in E2 in the Western blot is not clear and that is why we have increased the contrast of the original picture, so the band is more apparent. However, in terms of quantitation, and based on our experience, ELISA is a more reliable technique than a Western blot.

Other concerns are referring to the text, I don't see the point of including a prediction of a protein structure to confirm that it has the same structure of an identical protein: after codon optimization the aa sequence is the same so it is evident.

Answer: Agree. Figure 1 has been deleted.

The experiments analysing expression of genes involved in cytokinin signaling, homedomains, auxin signaling, under non-osmotic (normal) conditions and osmotic stress conditions can be a "plus" but I either don't see the point of including such an analysis in a paper where the main point is to get plant transformants in a easiest way. So, I think is quite a lot of work , and quite a lot of introduction , discussion and references focused on this matter and as I said before the objective of the paper should be the protocol of transformation and the effectivity of the plant for pest control.

Answer: Agree. All the information about the expression experiments and STRING analysis has been deleted, leaving only general information and a hypothesis of these genes’ involvement in the SE maturation.

Minor concerns:

Figure 6: define panels A and B in figure legend

Answer: Done.

Figure 7, spelling: transformation, not transformation

Answer: Done.

Figure 8: include a control with purified Cry10Aa

Answer: The main purpose of this analysis was the identification of the Cry10A protein in the transformed tissues rather than the comparison with the original protein or testing the accuracy of the technique. The identification of the Cry10A protein in the plant tissues was unquestionable, as each of the 17 detected peptides showed 100% identity with the reported Cry10A protein and all the peptides cover 76% of its sequence. This is an extremely high level of confidence. In mass spectrometry, it is considered that, if the independent peptides show 100% identity with the protein in question, 20% of coverage would be the minimum required to trust a reliable identification.

Reviewer 2 Report

Standardize in the manuscript – Cry10Aa not cry10Aa.

Fig. 1. Picture is marked as A, but there is only 1 picture.  

2.1.1. Generation of plasmid pMDC85/Cry10Aa – The Cry10Aa gene cloned in pUC57 – but  in M&M is pUC59 – so which one is it?

pMDC85 is 12462 bp, pMDC85/Cry10Aa gene 12828 bp according to legend of Fig. 2, but Cry10Aa gene is 1993 bp. So 12462+1993=14455 bp. Please verify this.

Fig. 2. The figure legend is not described well. All parts from scheme should be described in legend. There is lack of description of pVS1 StaA, pVS1 RepAm pVS1 ori. HygR is on scheme but in legend is hpt – please use the same abbreviation.

Coffea arabica, C. arabica, Arabidopsis thaliana – always italic.

Name of gene – always italic.

Table 1. Please use full name of genes or abbreviations. Some genes are up-regulated some are down-regulated, but You should add information how You calibrated expression.

Stable GFP expression was observed in 100% of transgenic lines – You mean 100% = 17 lines? Not 100%=835.

Table 2. CP2 – control but what kind of? In manuscript is also CP as negative control without transformation. CP2 and CP is something else?

How did You calculate a transformation efficiency?

SE-G medium – please describe this medium, there is no information.

qPCR analysis was performed to detect the presence of the Cry10Aa gene in the DNA – Why quantitative PCR? Regular PCR is enough for this. Please explain.

Only 12 of 17 transgenic lines had a Cry10Aa transgene, so why You analyzed the expression of Cry10Aa gene in all 17 transgenic lines? There no need to do that.

Fig. 4. In figure legend. All genes name should be in capitals.

ELISA and Western blot - Authors mentioned that somatic embryos were used for analysis. Did You prepare embryogenic culture from regenerants or select the somatic embryos during transformation and regeneration?

Fig. 8 – MWM?

In discussion

Somatic embryogenesis (SEG)? What G means?

Embryogenic calli at globular stage? – callus cannot be at globular stage.

Discussion part about SE-related genes is too long. There is only some package of information and lack of conclusions. Authors should focus on function of this genes in maturation of somatic embryos and effect of stress, and not put there all information about genes.

Author Response

Standardize in the manuscript – Cry10Aa not cry10Aa.

Answer: Done. Now all Cry10A proteins have been standardized with capital C, and all cry10A genes have been standardized with lower c and italics, as it should be.

Fig. 1. Picture is marked as A, but there is only 1 picture.

Answer: Figure 1 has been deleted, as suggested by another reviewer.

2.1.1. Generation of plasmid pMDC85/Cry10Aa – The Cry10Aa gene cloned in pUC57 – but in M&M is pUC59 – so which one is it?

Answer: It is pUC57. Correction has been made.

pMDC85 is 12462 bp, pMDC85/Cry10Aa gene 12828 bp according to legend of Fig. 2, but Cry10Aa gene is 1993 bp. So 12462+1993=14455 bp. Please verify this.

Answer: When we used the Gateway technology to clone the cry10A in the pMDC85 vector (12,462 bp), it interchanged the region between the attR2 and attR1 (1704 bp) by recombination, excising the sequence for the toxic protein ccdB, and integrating the cry10A gene cloned in TOPO8GWPCR. At the end, the vector pMDC85 plus the sequence for the toxic region of Cry10A is 12,838 bp long. The size is correct in the text.

Fig. 2. The figure legend is not described well. All parts from scheme should be described in legend. There is lack of description of pVS1 StaA, pVS1 RepAm pVS1 ori. HygR is on scheme but in legend is hpt – please use the same abbreviation.

Answer: Description of the plasmid has been thoroughly described in the figure foot.

Coffea arabica, C. arabica, Arabidopsis thaliana – always italic.

Answer: Corrections have been made.

Name of gene – always italic.

Answer: Corrections have been made.

Table 1. Please use full name of genes or abbreviations. Some genes are up-regulated some are down-regulated, but you should add information how you calibrated expression.

Answer: Table 1, along the related information in the text has been deleted, following the suggestion of another reviewer.

Stable GFP expression was observed in 100% of transgenic lines – You mean 100% = 17 lines? Not 100%=835.

Answer: It is from the 835 events. Clarification has been made in the text.

Table 2. CP2 – control but what kind of? In manuscript is also CP as negative control without transformation. CP2 and CP is something else?

Answer: CP2 is the basic embryo propagation medium (=SE-P) without any osmotic stress. This medium was used to prepare the different osmotic stresses by adding sucrose, mannitol, or sorbitol. CP is the pro-embryogenic mass induction medium (PEM), used to induce pro-embryogenic masses from the leaf explant. Some changes were made in the text to clarify this.

How did you calculate a transformation efficiency?

Answer: That is the proportion from the total bombarded embryogenic calli that grew in hygromycin medium. For example, from 50 plate containing 20 embryogenic calli each, 85 embryogenic events grew in hygromycin medium without osmotic stress agents (CP2). Text was modified to clarify this subject.

SE-G medium – please describe this medium, there is no information.

Answer: We are confused. In the section “Regeneration of transgenic plants” in Materials and Methods the SE-G medium is thoroughly described.

qPCR analysis was performed to detect the presence of the Cry10Aa gene in the DNA – Why quantitative PCR? Regular PCR is enough for this. Please explain.

Answer: Yes, that is correct. However, qPCR did not only prove the integration of the cry10A gene but also added quantitative data about the level of post-transcriptional expression for each transformation event. Those results were important to select transformed lines.

Only 12 of 17 transgenic lines had a Cry10Aa transgene, so why you analyzed the expression of Cry10Aa gene in all 17 transgenic lines? There no need to do that.

Answer: We included all 17 lines just to validate the results obtained by the PCR analysis. The results were corroborated.

Fig. 4. In figure legend. All genes name should be in capitals.

Answer: Figure 4, along the information related in the text has been deleted, following the suggestion of another reviewer.

ELISA and Western blot - Authors mentioned that somatic embryos were used for analysis. Did You prepare embryogenic culture from regenerants or select the somatic embryos during transformation and regeneration?

Answer: Analyzed somatic embryos were selected from established selected lines. Text was modified to make this clear.

Fig. 8 – MWM?

Answer: Correction has been done.

In discussion

Somatic embryogenesis (SEG)? What G means?

Answer: As mentioned in the sections “Abbreviations”, SEG is used for “somatic embryogenesis”, to differentiate it from SE for “somatic embryos”.

Embryogenic calli at globular stage? – callus cannot be at globular stage.

Answer: According to our understanding, an EMBRYOGENIC callus can be in globular stage. In any case, we changed that statement by using instead the term “embryogenic aggregate” in the text.

Discussion part about SE-related genes is too long. There is only some package of information and lack of conclusions.

Answer: That section has been deleted as suggested by another reviewer.

Authors should focus on function of this genes in maturation of somatic embryos and effect of stress, and not put there all information about genes.

Answer: That is exactly what we did. We deleted all this information, which is now added into another MS in preparation, only about this subject.

Reviewer 3 Report

The manuscript describes an efficient transformation of Coffea arabica somatic embryos through biolistic, where optimize the plant regeneration efficiency increasing the gelrite content in the medium. The authors relate this medium modification with the upregulation of some genes involves in coffee embryo development. A part of this interesting observation, as result of the transformation experiment, the authors obtain several transgenic lines containing Cry10Aa, which codifies a toxin of Bacillus thuringiensis, that could be useful to control the coffee berry borer disease.

The manuscript is in general well writing and the experimentation is well designed and conducted; there are only a few considerations, mainly regarding the figures.

In Fig. 1 remove the reference or include it inside the previous parenthesis; if not, it is like the prediction of the toxin has been done for other authors.

In Fig.2, 106 line, should be “Cry10Aa, under 2XCaMV35S”

In Fig.6 should be explained what is in the gel A and what is in the gel B; although, in my opinion, the gel A, which I believe is the DNA digestion, is not necessary and could be deleted.

In Fig 7 the letters of the statistics test at plant level show not differences, although they look as if there is a different, ¿is that correct? Similar results at embryos level showed significant differences. How can be explained these different results?

In Fig 8, in line 259 there is a mistake in the lines names. Apart of this, the western blot is not clear enough and I cannot see the 77Kda band, could you give more contrast to the image?

In Materials and methods, in line 402 is missed the hormone of the SE-M9 medium, I suppose that it is BAP.

In the discussion, the phrase started in line 271 must be reviewed, something is missed and it makes no sense.

As I said below, the manuscript describes a good work; but, in base of the results obtained, in which the amount of toxin is lower in plant leaves than in somatic embryos, I miss an assay of the interaction of plant- pathogen to test the effectiveness of the transformation to control the disease. Nevertheless, the study linking the use of high osmolarity and cytokinins in the medium with the regulation of genes involved in embryo development at seed level is of high interest and the obtaining of transgenic lines expressing Cry10Aa are very promising.

Author Response

The manuscript describes an efficient transformation of Coffea arabica somatic embryos through biolistic, where optimize the plant regeneration efficiency increasing the gelrite content in the medium. The authors relate this medium modification with the upregulation of some genes involves in coffee embryo development. A part of this interesting observation, as result of the transformation experiment, the authors obtain several transgenic lines containing Cry10Aa, which codifies a toxin of Bacillus thuringiensis, that could be useful to control the coffee berry borer disease.

The manuscript is in general well writing and the experimentation is well designed and conducted; there are only a few considerations, mainly regarding the figures.

In Fig. 1 remove the reference or include it inside the previous parenthesis; if not, it is like the prediction of the toxin has been done for other authors.

Answer: Figure 1 has been removed as suggested by another reviewer.

In Fig.2, 106 line, should be “Cry10Aa, under 2XCaMV35S”

Answer: Correction has been made.

In Fig.6 should be explained what is in the gel A and what is in the gel B; although, in my opinion, the gel A, which I believe is the DNA digestion, is not necessary and could be deleted.

Answer: Correction has been made. After delivering about the deletion of panel A, we decided that it is important to show that digestion was obtained as expected.

In Fig 7 the letters of the statistics test at plant level show not differences, although they look as if there is a different, ¿is that correct? Similar results at embryos level showed significant differences. How can be explained these different results?

Answer: Here there is a confusion that we have already clarified. The statistical differences were obtained by comparing the expression levels of plants and embryos within the same transformation event. Therefore, we found no statistical difference between embryos and plants from the transformation event 1 (E1), but we found statistical difference between expression in embryos and plants from transformation events E2, E3, and E4.

In Fig 8, in line 259 there is a mistake in the lines names. Apart of this, the western blot is not clear enough and I cannot see the 77Kda band, could you give more contrast to the image?

Answer: Correction has been made. Also, a more contrasted picture has been exchanged from the original one, where the band looks clearer.

In Materials and methods, in line 402 is missed the hormone of the SE-M9 medium, I suppose that it is BAP.

Answer: You are right. It is BAP and now it was added to the formula.

In the discussion, the phrase started in line 271 must be reviewed, something is missed and it makes no sense.

Answer: Phrase has been changed, expecting to be clearer.

As I said below, the manuscript describes a good work; but, in base of the results obtained, in which the amount of toxin is lower in plant leaves than in somatic embryos, I miss an assay of the interaction of plant- pathogen to test the effectiveness of the transformation to control the disease.

Answer: The importance of this MS is focused on the high transformation efficiency we obtained and the confirmation of the stable expression of the transgene reported here, as earlier reports show lower transformation efficiency and regeneration of coffee plants. We understand that the MS would be significantly improved with the addition of efficiency data against CBB; however, as you know, coffee is a perennial plant, meaning that the first flowering occurs after 4 to 5 years after plantation. At this moment, we have transformed coffee trees about 1.5 meters tall and hundreds of smaller plants, and we expect to have the first flowering within 1 year. We expect to have an MS ready, about their insecticidal activity, within 2 years.

Nevertheless, the study linking the use of high osmolarity and cytokinins in the medium with the regulation of genes involved in embryo development at seed level is of high interest and the obtaining of transgenic lines expressing Cry10Aa are very promising.

Comment: Yes. That is exactly what we want to update now.

Round 2

Reviewer 1 Report

Authors have addressed most of the questions that were raised and I think that the paper could be published as it is now in the new version.